# Oscillating edge states in one-dimensional $MoS_2$ nanowires

Hai Xu[1,2], Shuanglong Liu[3], Zijing Ding[3,4], Sherman J. R. Tan[1,5], Kah Meng Yam[1,2], Yang Bao[1], Chang Tai Nai[1], Man-Fai Ng[6], Jiong Lu[1,2], Chun Zhang[1,3] & Kian Ping Loh[1,2]

Reducing the dimensionality of transition metal dichalcogenides to one dimension opens it to structural and electronic modulation related to charge density wave and quantum correlation effects arising from edge states. The greater flexibility of a molecular scale nanowire allows a strain-imposing substrate to exert structural and electronic modulation on it, leading to an interplay between the curvature-induced influences and intrinsic ground-state topology. Herein, the templated growth of $MoS_2$ nanowire arrays consisting of the smallest stoichiometric $MoS_2$ building blocks is investigated using scanning tunnelling microscopy and non-contact atomic force microscopy. Our results show that lattice strain imposed on a nanowire causes the energy of the edge states to oscillate periodically along its length in phase with the period of the substrate topographical modulation. This periodic oscillation vanishes when individual $MoS_2$ nanowires join to form a wider nanoribbon, revealing that the strain-induced modulation depends on in-plane rigidity, which increases with system size.

[1] Department of Chemistry, National University of Singapore, 3 Science Driver 3, Singapore 117543, Singapore. [2] Centre for Advanced 2D Materials and Graphene Research Centre, National University of Singapore, Singapore 117546, Singapore. [3] Department of Physics, National University of Singapore, Singapore 117551, Singapore. [4] SZU-NUS Collaborative Innovation Center for Optoelectronic Science and Technology, College of Optoelectronic Engineering, Shenzhen University, Shenzhen 518060, China. [5] NUS Graduate School for Integrative Sciences and Engineering, National University of Singapore , Centre for Life Sciences, #05-01, 28 Medical Drive, Singapore 117456 Singapore. [6] Institute of High Performance Computing, Agency for Science, Technology and Research, 1 Fusionopolis Way #16-16 Connexis, Singapore 138632, Singapore. Correspondence and requests for materials should be addressed to K.P.L. (email: chmlohkp@nus.edu.sg).

Low-dimensional transition metal dichlacogenide nano-structures display numerous intriguing physical and chemical phenomena, owing to the presence of strong electron confinement, boundary states and edge polarization[1–7]. For instance, a one-dimensional (1D) $MoS_2$ nanoribbon is predicted to possess novel properties such as metallic edge states, 1D confined plasmons and ferromagnetic behaviors depending on its width and edge configurations[8–11]. When the dimensionality is reduced further to single molecular-scale wire constructed from hexagonal repeating motifs of Mo–S bonds, the electronic states are predicted to be further re-normalized due to the electron–phonon coupling and electron–electron interactions, leading to exotic electronic superlattice and magnetic ordering[2,12,13]. The 1D nanowire's width is defined by its molecular unit cell and presents two fundamental differences compared with a two-dimensional (2D) system. First, owing to the narrow width of such 1D nanowires, its structure is highly pliable and is open to modulation by topographical strain. Curvature effects on a quantum nanowire with Rashba spin–orbit coupling have been demonstrated to promote the generation of non-trivial edge states and topological insulating phase[14,15]. Other properties predicted for curvature-induced electronic and transport properties include winding-generated bound states and large anisotropic magnetoresistance[3,16]. Second, the lack of inversion symmetry for $MoS_2$ edges lead to a rich variety of edge structures[5,17–20], which can have pronounced influences on the electronic properties of the nanowires. Recently, $Mo_6S_6$ nanowires have been successfully fabricated and characterized using focus electron beam lithography[4]. However, top-down methods are disadvantaged by the lack of structural control and scalable production. In contrast to top-down approaches, bottom-up self-assembly method affords accurate control of size and edge structures, as has been demonstrated for the growth of 1D graphene nanoribbons[21]. Bottom-up self-assembly can be initiated by atom evaporation on anisotropic substrates, where the broken symmetry can be created by the presence of step edges. At the initial stage of 1D growth of atomic chains, strong dipole–dipole repulsion may discourage the assemblies of atomic wires and promote the formation of hexagonal layers, unless these are stabilized by the substrate. To date, the synthesis of stoichiometric $MoS_2$ nanowires, distinct from 1D inversion domain boundaries, has not been reported and their fundamental electronic properties remain largely unexplored.

Herein, we apply a surface-templated approach to fabricate and precisely align $MoS_2$ nanowires (width < 1 nm) and nanoribbons (width > 1 nm). In-situ scanning tunnelling microscopy (STM) and non-contact atomic force microscope (NC-AFM) imaging are employed to determine the topographic corrugation and electronic structures of the nanowires at the atomic scale. The transition from 1D $MoS_2$ wires to wider 1D nanoribbon provides an excellent opportunity to study the lateral fusion of wires and the relationship between strain and system size.

## Results

**1D $MoS_2$ nanowires growth.** Owing to its broken symmetry, regular steps or periodically faceted surfaces have been used as templates for the organized growth of low-dimensional nanostructures (Fig. 1a). On the Au (111) surface, following the sequential evaporation of S and Mo atoms, sparsely distributed $MoS_2$ wires are seen to align predominantly along step edges (Fig. 1b) and there are competitive growth of hexagonal islands and clusters on the terraces at higher temperatures (Supplementary Fig. 1). To achieve 1D-templated growth, we select Au (755) surface, which can be described as a reconstruction-free, atomic staircase consisting of 1.42 nm-wide

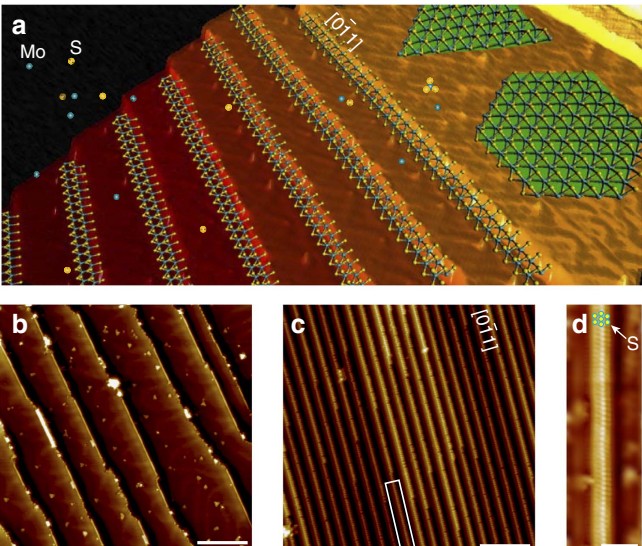

**Figure 1 | Bottom-up synthesis of $MoS_2$ nanowire array on the vicinal Au surface.** (**a**) Schematic illustration of the step-templated growth of $MoS_2$ nanowires on the Au(755) surface. (**b**) Large-area STM image of $MoS_2$ triangular nanoclusters and $MoS_2$ nanowires grown on an Au(111) surface. (**c**) Large-area STM image of $MoS_2$ nanowires aligned along the steps on the Au(755) surface. (**d**) Magnified view of the rectangle-enclosed region in **c** shows the atomic-resolved lattice of a single $MoS_2$ nanowire (S atom positions are indicated by yellow circles). Scale bars, 10 nm in (**b,c**) and 1 nm (**d**).

terraces separated by monatomic {100} steps[22,23] (Fig. 1a and Supplementary Discussion). By sequential evaporation of the binary Mo and S elements, a high density of well-aligned, ~200 nm-long $MoS_2$ nanowires can be grown on the steps of Au (755) between 150 °C and 250 °C, as shown in the large-scale STM image (Fig. 1c). The sequential structural evolution of the nanowire from its elemental precursors to the formation of a binary compound has been carefully tracked using STM (Supplementary Figs 1–3), which reveals that $MoS_2$ nanowire has markedly different morphologies from self-assembled structures based on the pure elemental sources of only S or Mo. Figure 1d shows a high-resolution STM image of the wires, which can be resolved as three bright chains within a width of ~0.7 nm. We also performed high-resolution frequency modulation NC-AFM imaging, to elucidate the atomic structure and the topographic corrugation of the $MoS_2$ nanowires. As shown in Fig. 2a,b, the topography and its derivative NC-AFM images of a single $MoS_2$ nanowire exhibits three chains of bright spots similar to the STM image (Fig. 1c) with a width of ~0.67 nm (bright spots correspond to the S atomic positions in $MoS_2$ nanowires, also see Fig. 2f). The average lattice spacing along the axis of nanowires is measured to be ~2.96 Å (Fig. 2e), which is in-between the lattice constants of Au(111) (2.88 Å) and bulk $MoS_2$ (3.15 Å), suggesting that the $MoS_2$ nanowires are under compressive stress.

The chemical bonding in these nanowires has been analysed using X-ray photoelectron spectroscopy (XPS) and high-resolution electron energy loss spectroscopy (HREELS; Supplementary Figs 4 and 5, and Supplementary Table 1). The binding energies of the Mo $3d_{5/2}$ state is at 228.65 eV, which is chemically shifted to lower energy compared with $MoS_2$ islands grown on Au(111); this can be explained by charge transfer from gold, which increases the electron density in the nanowire. HREELS analysis reveals three peaks in the $MoS_2$ nanowire samples. The first peak located at 49.0 meV matches the Mo–S stretching mode in the $MoS_2$ trigonal prismatic coordinated

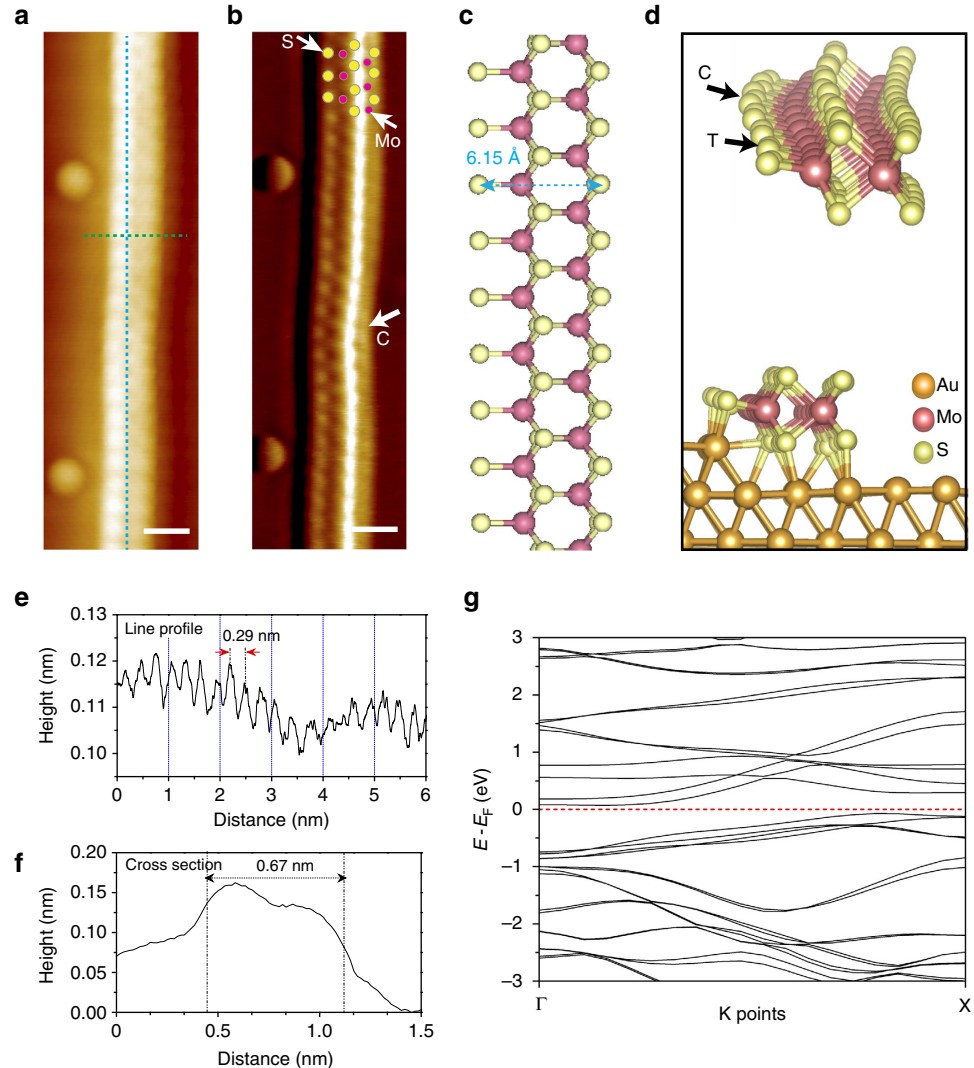

**Figure 2 | Atomic structure of a single MoS$_2$ nanowire grown on the Au (755) surface. (a)** High-resolution NC-AFM image of a single MoS$_2$ nanowire reveals the lateral topographic modulation with a period of 4.4 nm ($f_0 = 23.1$ kHz, $df = -16$ Hz, $T = 4.5$ K). **(b)** The corresponding derivative image of **a** resolves three atomic chains for a single MoS$_2$ nanowire. Mo(S) atom positions are indicated by rose (yellow) circles. In a laterally corrugated nanowire, the region closest to (furthest away from) the step edge is defined as crest, C (trough, T) site as marked by white arrows. **(c)** Schematic illustration of the atomic structure of a single MoS$_2$ nanowire (yellow ball: S, rose ball: Mo). **(d)** DFT simulation of a single MoS$_2$ nanowire grown along the step edge of Au (755) surface (gold ball: Au atom). **(e)** The line profile obtained along the axis of a single MoS$_2$ wire reveals a compressed lattice spacing of 0.296 nm. **(f)** The line profile across the MoS$_2$ nanowire. **(g)** Calculated band structure of a single suspended MoS$_2$ nanowire. Scale bars, 0.5 nm in all NC-AFM images.

structure. We assign the 98.0 and 135.5 meV peaks to the Mo–S edge atoms on both asymmetric ends of the MoS$_2$ nanowires (Supplementary Discussion).

Based on these observations, we propose that the nanowire should be constructed from repeating hexagonal MoS$_2$ units with trigonal prismatic coordination and it has zigzag Mo edges terminated by S atoms, as illustrated in Fig. 2c,d. The termination of the Mo edge atoms by S atoms is based on the principle that such mono-S terminated Mo edges are energetically stable in a sulfur-rich environment[8,18,24]. The skeleton of the proposed nanowire consists of three parallel rows of atoms, which fits the three rows of atoms observed in STM and NC-AFM. In this atomic model, all S atoms are situated at the hollow sites of Au(111), yielding a compressively stressed lattice constant of 2.96 Å (~3% strain) for the MoS$_2$ wire along the [0$\bar{1}$1] direction. The width of the proposed nanowire, counting from one sulfur edge to another, is 0.615 nm, which agrees with our NC-AFM measurements.

**The electronic structure of 1D MoS$_2$ nanowires.** Interestingly, a long-range lateral topographic corrugation with a period of 4.4 ± 0.2 nm developed along the length of the wire. As such, corrugations are not observed on edges or domain boundaries of larger MoS$_2$ islands, we speculate that it may be related to 1D superlattice induced by the compressive stress when the wire aligns itself with the Au (111) lattice along the [0$\bar{1}$1] direction. It is noted that the periodicity of this corrugation is much larger than the 2D moiré pattern formed by MoS$_2$ monolayer film on Au(111) surface, which has a typical in-plane periodicity of 3.2 ± 0.2 nm (ref. 25). The large period of lateral superlattice observed here is mainly attributed to the chemical bonding between edge S and Au atoms at the step edge, which will be discussed later. The supra-atomic periodicity at the edge of MoS$_2$ nanowires follows a unique (6, 8) pattern, where six atoms closer to step edges are involved in one period (defined as crest region) and eight atoms away from step edges form another period (defined as trough region), leading to a periodic topographic modulation.

The periodic topographical corrugation is expected to modulate the local electronic properties of $MoS_2$ nanowires in a periodic manner. To explore this effect, we conducted spatially resolved scanning tunnelling spectroscopy (STS) measurement, where $dI/dV$ spectra reflect the energy-resolved local density of states (LDOS). Figure 3a shows a set of $dI/dV$ spectra obtained at different positions over one modulation period along the axis of the $MoS_2$ nanowire (from point A to point B as indicated in the inset of Fig. 3a). In the positive sample bias regime, the $dI/dV$ spectra are almost featureless, except for a small rising shoulder appearing at $0.75 \pm 0.05$ eV above the Fermi level ($E_F$), labelled as $P_1$, which is a typical peak on $MoS_2$/Au system, corresponding to the onset of $MoS_2$ conducting band[25]. In the negative sample bias regime, two prominent features marked as $P_2$ ($-0.6 \pm 0.05$ eV) and $P_3$ ($-1.0 \pm 0.05$ eV) emerge in the spectra and oscillate spatially along the axis of nanowires. Although the position of $P_3$ appears to match the valence band maximum, we did not find an origin for $P_2$ arising from any hybridized orbitals or energy states within the basal plane of large sized $MoS_2$ islands. As our $MoS_2$ nanowires are essentially made of closely spaced edges, together with the observation of significant LDOS intensity along the step-contacted edge of the nanowires in the $dI/dV$ mapping of these two peaks, we deduce that $P_2$ and $P_3$ originate from the edge states on the $MoS_2$ nanowires. $P_2$ is dominating in the crest region

(close to the step edge) but suppressed in the trough region (away from the step edge). In contrast, $P_3$ is dominating in the trough region but vanishes in the crest region. Hence, $P_2$ and $P_3$ are out of phase with each other and alternate with a $4.4 \pm 0.2$ nm period that follows the topographical modulation (Fig. 3b). In addition, we also carried out the $dI/dV$ mapping at a bias voltage of $-1.0$ V (Fig. 3c) and $-0.6$ V (Fig. 3d) to reveal the spatial distribution of $P_2$ and $P_3$ states, respectively. In both cases, the local differential conductance fluctuates periodically along the wire's length, such that the bright and dark regions alternate with a period consistent with the topographical modulation of $4.4 \pm 0.2$ nm. The $dI/dV$ map acquired at $-1.0$ V reveals an oscillation of the local differential conductance opposite to that acquired at a bias voltage of $-0.6$ V (Fig. 3d). Near the Fermi level, higher LDOS intensity shifts to the interior regions from the edges. In addition, asymmetry of the LDOS distribution was observed around $E_F$ in $dI/dV$ mapping (Fig. 3e,f). This asymmetric behaviour is similar to the moiré pattern inversion between the filled and empty electronic states in monolayer graphene/Ru system[26]. Close-up image shows that each intensity maxima follows a (6, 8) pattern where six atoms are involved in one period, followed by another eight atoms, consistent with periodicity observed in our NC-AFM imaging.

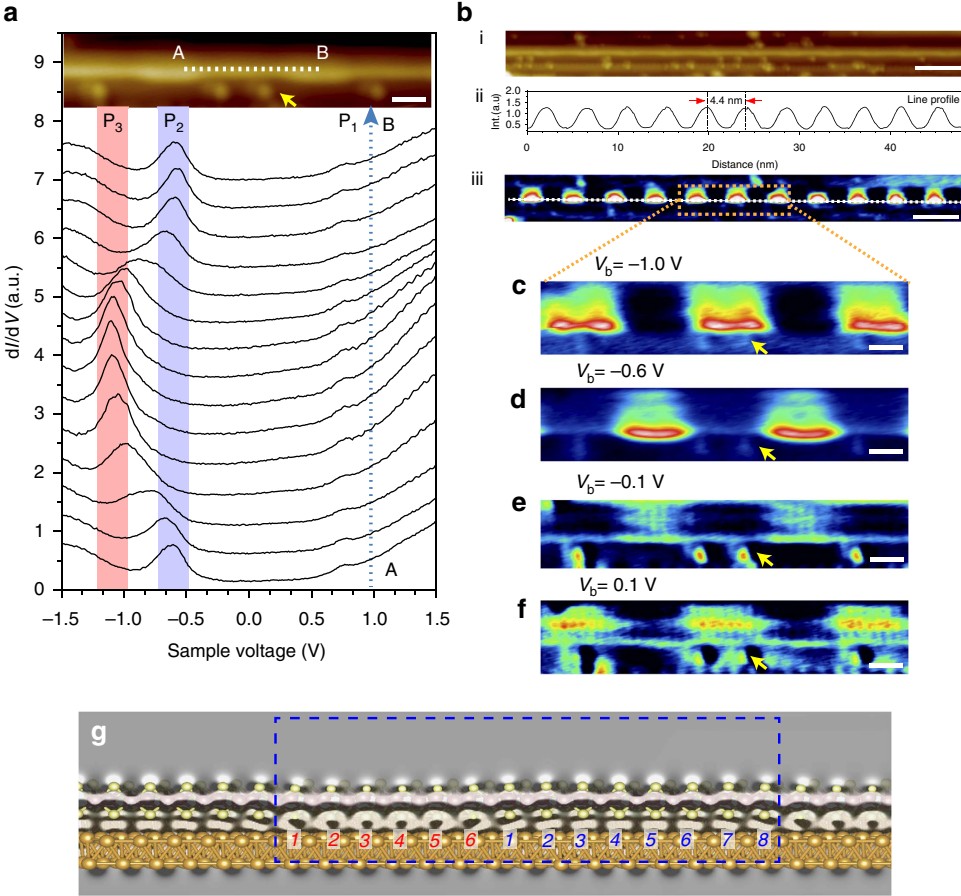

**Figure 3 | Oscillating electronic structures in a single $MoS_2$ nanowire. (a)** A set of spatially-resolved STS spectra acquired along the axis of a single $MoS_2$ nanowire over one modulation periodicity (from A to B marked in the inset STM image, $V_b = -0.6$ V, $I_t = 150$ pA ). **(b**, i) STM image and (iii) corresponding $dI/dV$ mapping of an extended single $MoS_2$ nanowire ($V_b = -1.0$ V, $I_t = 150$ pA); (ii) line profile along the single $MoS_2$ nanowire in $dI/dV$ map shows a periodic electronic modulation. **(c-e)** Experimental $dI/dV$ maps of the rectangle-enclosed region in **b**,iii taken at different sample bias: **(c)** $-1.0$ V, **(d)** $-0.6$ V, **(e)** $-0.1$ Vand **(f)** 0.1 V . The same position is marked by yellow arrows in STM image and corresponding $dI/dV$ maps. **(g)** DFT calculation of the charge distribution in the single $MoS_2$ nanowire reveals the (6,8) pattern as highlighted by red and blue numbers. Blue dotted box demarcates a single modulation period. Scale bars, 5 nm (**b**) and 1 nm (**a**,**c-e**).

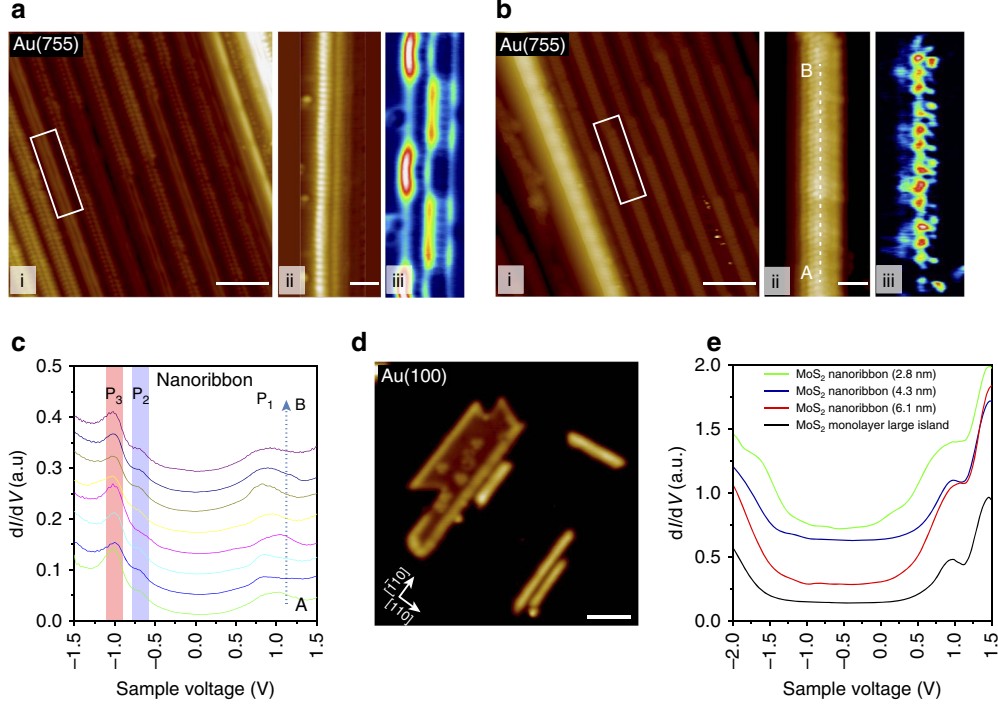

**Figure 4 | Electronic structure characterization of twin nanowires and MoS$_2$ nanoribbons.** (**a**, i) Large-scale STM image of twin nanowire arrays grown on the Au (755) surface and (ii) the close up STM image as well as (iii) the corresponding d$I$/d$V$ map obtained at $V_b = -0.6$ V in the right panel. (**b**, i) Large-area STM image of the MoS$_2$ nanoribbons array and (ii) the close up STM image as well as (iii) the corresponding d$I$/d$V$ map obtained at $V_b = -0.6$ V in the right panel. (**c**) A set of spatially resolved d$I$/d$V$ spectra taken along the long axis of MoS$_2$ nanoribbons as marked from position A to position B in the STM image (**b**,**d**), MoS$_2$ nanoribbons with different sizes synthesized on Au (100) surface. (**e**) d$I$/d$V$ spectra acquired at the center of a series of MoS$_2$ nanoribbons show the size-dependent bandgap. Scale bars, 5 nm (**a**,**b**), 1 nm for close up images and 10 nm (**d**) respectively.

**Density-functional theory calculation.** To provide insight into the electronic properties of the system, we performed first-principles density-functional theory (DFT) calculations (with the VASP package[27]) on ground-state atomic and electronic structures of a single MoS$_2$ wire on Au (755) surface. DFT calculations were performed with a plane wave basis (350 eV for the kinetic energy cutoff). In all calculations, the local spin density approximation and the generalized gradient approximation in Perdew–Burke–Ernzerhof format[28] together with the projector augmented wave method[29] were included. In optimizing atomic structures, the force convergence criterion was set to 0.01 eV Å$^{-1}$. For a single suspended MoS$_2$ nanowire, our calculations showed that the lattice constant of the wire is around 3.15 Å. The lattice constants of suspended MoS$_2$ nanowire and Au(111) are 3.15 and 2.88 Å, respectively; thus, there is significant lattice mismatch of ~9% (ref. 25). To minimize the effects of the mechanical strain, we chose a supercell model that contains 14 units of the MoS$_2$ nanowire on 15 units of the Au step edge, to reduce the lattice mismatch between MoS$_2$ and Au substrate to ~3%. The optimized adsorption geometry is shown in Fig. 2c,d from which we can clearly see that when supported on Au (755), the wire corrugates along the direction parallel to the Au surface to release the mechanical strain. The charge transfer between the Au substrate and the wire was calculated using the Bader charge analysis, demonstrating that Au surface loses electrons and the S atoms in the wire gain electrons. The isosurface of the charge redistribution is shown in Fig. 3g, which clearly shows the electronic structure oscillation with the (6, 8) pattern in one supercell that agrees well with the experiment observation. More detailed analysis from the DOS projected on the wire that is supported on Au (Supplementary Fig. 6) shows two significant states at around −0.6 and −1.0 eV (from GS

calculation). DFT calculation shows that the S atoms that form bonds with Au atoms both on the step and the surface gain the most electrons (> 0.5 $e$; Supplementary Fig. 6). The non-uniform binding interaction between these S atoms and the Au substrate is the main driving force, leading to the topographical modulation and edge-state oscillation of the MoS$_2$ nanowire. In addition, our calculations show that the suspended nanowire is a semiconductor with a small band gap of 0.14 eV (Fig. 2g and Supplementary Discussion), whereas the supported nanowire becomes metallic arising from its strong hybridization with the Au substrate. It is worth mentioning here that STM tip-induced local field should exert significant effects on the electronic properties of these ultrathin wires, and to investigate these effects first-principles theory for non-equilibrium quantum systems such as the recently proposed steady-state DFT[30] is needed. We will address this in our future studies.

The 1D nanowires show an ability to self-assemble in a row by row manner. When the concentration of the atoms on the surface is increased, the Mo edge of the as-grown single wire on a step acts as a 'nucleation edge' for the assembly of an adjacent wire on the upper terrace, such that a twin MoS$_2$ nanowire is produced, shown in Fig. 4a (see Supplementary Fig. 7). The width of twin wire is measured to be 1.40 ± 0.1 nm. The zig-zag edges of two MoS$_2$ wires can share Mo or S edge atoms, forming a joined-edge defect identified by the presence of a 'dark' line in the STM image in Fig. 4a. This is analogous to the sulfur 'mirror boundary' or inversion domain boundary observed in the boundary of MoSe$_2$ islands[31,32]. At this stage, this twin wire structure still exhibits the periodic topographic corrugation and the oscillation of the edge state, similar to the single wire. As shown in Fig. 5, individual nanowires in this twin structure exhibit an unexpected out-of-phase modulation of the edge states as revealed in d$I$/d$V$ mapping

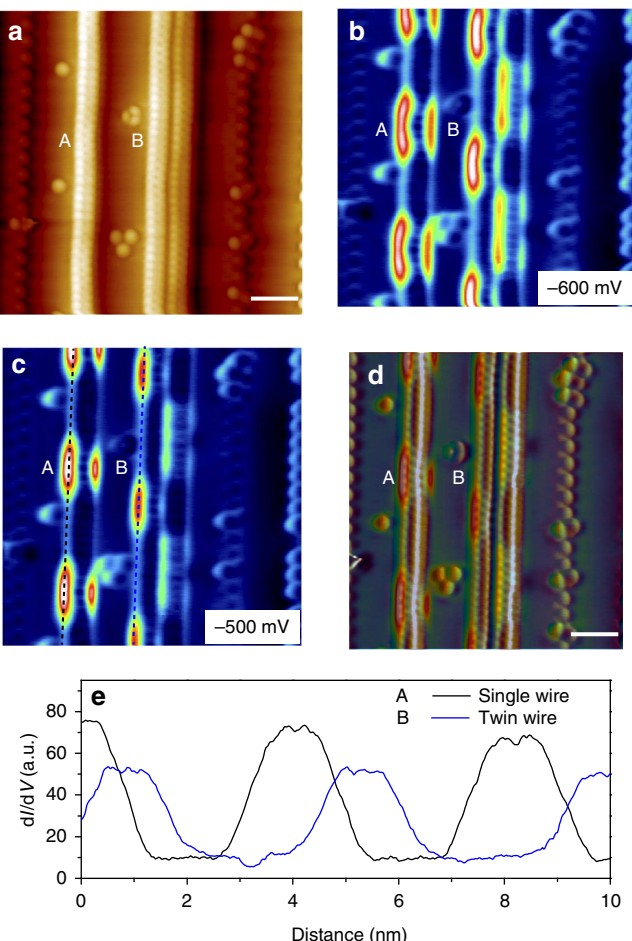

**Figure 5 | Periodic electronic patterns on the MoS₂ nanowires.**
(**a**) Topography image of MoS$_2$ nanowires obtained by high-resolution NC-AFM ($f_0 = 23.1$ kHz, $df = -16$ Hz, $T = 4.5$ K). What is shown here includes a single nanowire (A) and a twin nanowire (B). The dark line between the twin nanowires suggests that the individual wire has not fused and there is coulombic repulsion between the two wires. (**b,c**) Typical d$I$/d$V$ maps (10 × 10 nm$^2$) recorded in at various bias voltage with same tip with **a** in STM mode. (**d**) d$I$/d$V$ map ($V_b = -600$ mV) overlaid onto the NC-AFM topographic image. (**e**) Line profile along the MoS$_2$ wires in **c** denoted by black (blue) dotted lines, demonstrating the same periodic oscillation of tunnelling conductance along them. Scale bars, 1 nm (**a,d**).

acquired at $-0.6$ eV, even though they adopt an in-phase periodic topographic corrugation. This observation suggests that edge states in the twin wire structure are coupled with each other through electronic correlation effect arising from Coulomb interactions, which further modulates their electronic structure. When the growth temperature is increased to 250 °C, the twin wire transforms into a seamlessly fused nanoribbon, as can be judged by the vanishing of the joined-edge defect in the closed-up STM image (see magnified image of twin wire in Fig. 4a and nanoribbon in Fig. 4b). Interestingly, the topographic corrugation has disappeared in this nanoribbon and the edges become straight and parallel to the Au {100} steps. Spatially resolved STS in Fig. 4c show that the periodic fluctuations of the edge state observed in MoS$_2$ nanowires have vanished in the nanoribbon, although the edge states still exist. This suggests that the interaction of the wider ribbon with the gold surface is much weaker than the single wire, presumably due to the higher in-plane stiffness when the system size increases. As the population of the basal plane atoms

increase relative to that of the edge atoms, the increased interlayer and intralayer Mo–S charge transfer increases the in-plane rigidity and resists lateral distortion by the gold lattice.

## Discussion

To confirm whether the substrate-induced strain of the MoS$_2$ nanowire by the Au substrate is mitigated when the system size increases, we select Au (100) surface for the growth of wider MoS$_2$ ribbons. The Au (100) substrate has fourfold symmetry and templates the growth of wider MoS$_2$ nanoribbons in biaxial <110> directions. Figure 4d shows the growth of MoS$_2$ nanoribbon along two <110> directions with a 90° intersection angle on the Au(100) surface. The d$I$/d$V$ spectra recorded on the ribbons as a function of widths are shown in Fig. 4e, where the STS-measured band gap gradually increases with the ribbon width. When the nanoribbon reaches 6 nm in width, the bandgap is about 1.5 eV, which is close to the bulk value of 1.7 eV obtained for MoS$_2$/Au (111) system[19,25]. The periodic oscillation of the edge states has vanished in wider ribbons (Supplementary Fig. 8), which indicates very clearly that the edge corrugation effects become increasingly weaker due to the increased in-plane rigidity of the system. We have also studied the edges and grain boundaries of larger 2D MoS$_2$ islands and did not observe periodic oscillations of the DOS states or edge-related states (Supplementary Fig. 8 and Supplementary Discussion). Basically, all DOS peaks exhibit spatially constant intensity in the tunnelling current throughout. The lateral corrugation of the 1D MoS$_2$ nanowire observed here is distinct from the 2D moiré pattern observed for the 2D MoS$_2$ monolayer on Au (111); the latter arises due to 2D compressive stress and the very low out-of-plane bending rigidity of the 2D monolayer. In the case of the 1D MoS$_2$ wire, owing to the finite size of the molecular scale system and asymmetrical bonding of S edge atoms to the Au step edge (only one side is bonded), it is readily laterally corrugated, producing a 1D corrugation period that is much larger than the 2D moiré lattice. As adjacent MoS$_2$ nanowires joined across the steps, the bonding to the step edge is replaced by in-plane Mo–S bonding, thus lifting the substrate-induced lateral corrugation and the edge-state oscillation disappears. For larger 2D MoS$_2$ system, the in-plane rigidity has increased significantly compared with the out-of-plane; hence, the corrugation is preferred in the vertical direction.

In summary, our studies show that due to the pliability of the single MoS$_2$ nanowire, it undergoes strong coupling with the Au substrate, giving rise to a substrate-modulated superlattice potential. This coupling follows a (6, 8) long-range pattern where six atoms in one section and eight atoms in an adjacent section forms a 1D wave due to varying degrees of bonding interactions with Au, with simultaneous periodic modulation on its electronic structures. In phase with the (6, 8) pattern, the edge states oscillate between $-0.6$ and $-1.0$ eV along the length of the wire, giving rise to a unique bias-dependent charge-ordering effects. The existence of the oscillating edge states potentially generates well-defined periodic transmission channels for studying coherent transport and quantum interference in low-dimensional nanodevices. If a magnetic field is applied perpendicularly to the nanowires, an array of the transmission channels in 1D nanowires can serve as uniformly spaced electron traversing pathways for constructing an Aharonov–Bohm interferometer[33,34]. Parallel nanowires exhibit an 'in-phase' charge ordering if the separation distance is larger than the terrace width, whereas adjacent wires show an 'out-of-phase' charge reordering if the separation is reduced to an atomic-bond level, presumably due to the enhanced Coulomb interactions (Fig. 5). It has been predicted by theoretical calculations that

zigzag $MoS_2$ nanoribbons become half-metallic as a result of the $(2 \times 1)$ reconstruction of edge atoms and are semiconductor for minority spins, but metallic for the majority spins[35]. Using first-principles calculations, Chu *et. al.*[20] predicted that chalcogen-terminated zigzag edges support edge bands with strong Rashba-type spin–orbit coupling, which are well separated from the bulk bands; the edge modes can be topological, although the bulk semiconductor is non-topological. These effects are expected to be accentuated in transition metal dichalcogenides with large spin–orbit couplings. The potential modulation and charge ordering observed here could result in the peculiar spin texture and coherent spin dynamics for spin manipulation and detection in these wires.

## Methods

**Sample preparation.** Experiments were conducted in ultra-high vacuum (UHV) condition. Monolayer $MoS_2$ islands, nanoribbons and $MoS_2$ wires were grown by molecular beam epitaxy on Au surfaces in prepare chamber. Mo was deposited from an e-beam evaporator (EF3, Omicron) on Au surfaces at $\sim 100\,°C$, while maintaining the pressure at $3.0 \times 10^{-10}$ Torr, followed by annealing the substrates at 150–500 °C in a sequence of sulfur-rich environment through exposure to $H_2S$ at $1 \times 10^{-6}$ Torr.

**STM and NC-AFM imaging.** After the sample growth, the sample was transferred into the analysis chamber, and STM/NC-AFM imaging and STS measurements were performed at $T = 4.5\,K$. STS differential conductance $(dI/dV)$ point spectra and spatial maps were measured in constant-height mode using standard lock-in techniques $(f = 773.1\,Hz, V_{r.m.s.} = 16\,mV, T = 4.5\,K)$. $dI/dV$ spectra on Au(111) substrate was used as an STS reference for tip calibration. NC-AFM images were recorded in a constant frequency mode (sensor frequency $f_0 \approx 23\,kHz, Q \approx 20,000$). NC-AFM images were measured at a sample bias $V_b = -40\,mV$. STM/STS data were analysed and rendered using SPIP software.

**XPS and HREELS analysis.** XPS analysis was performed with a Mg Kα emission line (1253 eV) using a XR 50 X-ray source (SPECS GmbH). The photoelectrons were analysed by a PHOIBOS 150 hemispherical analyser (SPECS GmbH), working with a 30 eV pass energy. The binding energy was calibrated by the Au 4f reference (Au $4f_{7/2} = 84.0\,eV$). HREELS was performed using a Delta 0.5 spectrometer (SPECS GmbH) in a UHV chamber with a base pressure of $2 \times 10^{-10}$ mbar with an incident energy of 4 eV. The specular scattered electrons were collected and counted by a Galileo 4830 U channeltron. To protect samples from contamination and oxidation during transport through air to the UHV-XPS and HREELS chambers, a home-made vacuum transporter was used to transfer the samples between the analysis system and STM UHV systems.

**Data availability.** The authors declare that the data supporting the findings of this study are available within the article and its Supplementary Information files.

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

## Acknowledgements

We acknowledge support from the National Research Foundation (NRF), Prime Minister's Office, Singapore, under its Medium Sized Centre (CA2DM) programme.

## Author contributions

H.X. and K.P.L. conceived the project. H.X. and Y.B. performed $MoS_2$ sample growth and STM characterization. S.L.L., Z.J.D., K.M.Y. and M.-F.N. contributed to the DFT calculations. S.J.R.T. and C.T.N. performed XPS and HREELS measurements. H.X. and K.P.L. wrote the paper with contributions from C.T.N., J.L. and C.Z., and C.T.N. supplied the illustrations. All authors contributed to the data analysis and read the manuscript.

## Additional information

**Competing financial interests:** The authors declare no competing financial interests.

