## [Peer Review File · Nature Communications]

Reviewers' comments:

Reviewer #1 (Remarks to the Author):

The manuscript reports the preparation of MoS₂ nanowire and nanoribbon on a vicinal Au substrate. The strain induced morphology corrugation and associated edge state modulation were revealed very clearly by using STM/STS combined with first principles calculations. The data and analysis are clean and convincing. However I feel that the reported result is conceptually not new, and not so important to stimulate a general interest. Edge state is commonly observed on surface steps and 1D structures, and strain modulation due to lattice mismatch is very natural. It is not surprising to observe such phenomena. The authors claim that such modulation along 1D wire may tune the 1D transport property, which is of course interesting if the authors can demonstrate it. But that is totally another story. For the present work, I feel it more suitable for a more specific journal such as physical review B.

Reviewer #2 (Remarks to the Author):

The manuscript reports the synthesis and characterization of different new MoS₂ nanowires (around 200 nm long) grown by molecular beam epitaxy using gold substrates. The nanowires/nonoribbons grow at the steps of Au (755) surface. In particular, the thinnest nanowire exhibits a structural and electronic modulation due to the edge interaction with the Au substrate. As the nanowires get wider, the modulation vanishes since the interaction with the substrate changes. Different characterization techniques are used to analyze the properties of these MoS₂ nanowires such as STM, NC-AFM, XPS and HREELS. Additionally, the authors have performed first principles calculations to support their findings.

The results look correct, interesting, useful to experimentalists and theorists, and could lead to the controlled synthesis of other semiconducting transition metal dichalcogenides (STMD) nanowires/nanoribbons, similar in structure (trigonal prismatic), such as WS₂, MoSe₂ and WSe₂. I recommend the publication of the manuscript after the authors address the following points:

- 1.- How stable would the MoS₂ nanowires be if these are exposed to air?
- 2.- Can the authors comment about the possibility of releasing the nanowires from the gold substrate to use them in other applications? Is this possible?
- 3.- According to the authors DFT calculations, the suspended thinnest nanowire possesses a band gap of 0.14 eV. Since DFT always underestimates the band gap, the authors should mention this in the text. Can the authors provide a theoretical realistic value of this band gap for the suspended system?
- 4.- Can the authors comment, in the context of their findings, how their technique could be extended to other semiconducting STMD such as WS₂ and WSe₂ and how the spin orbit coupling, which is greater

than in MoS₂, could play a role in new electronic/magnetic properties, when the nanoribbons get wider?

Reviewer #3 (Remarks to the Author):

Mainly with the STM, Authors studied one-dimensional MoS₂ grown at the step edge of Au surface, found a periodical oscillation. The observation is original and interest, although this phenomenon is very similar to 2D moire pattern.

the data were collected carefully and analyzed reasonable, the presentation is clear, the conclusions sound right and useful.

There are a few points should be considered before publication.

1. the treatment of uncertainties. Authors mentioned the period is 4.4 ± 0.2 nm, but I cannot find why is ± 0.2 , please specify it.
2. Line 151,152. "On the other hand, it is noted that a peak with similar position as P2 has been reported on the edges of MoSe₂". I do not understand why authors mentioned a MoSe₂ peak. The materials are different; it is useless to compare two STS peaks from two materials.
3. Line 168, 169. "This asymmetric behaviour is similar to the moire pattern inversion between the filled and empty electronic states in monolayer MoS₂/Au system." Please give a Ref. article.
4. In this manuscript, Line 118, the edge of MoS₂ was considered be terminated by mono-S. Do Authors check the edge be terminated by dimer-S?
5. Based on the STM image and DFT calculation, author show the structures of the nanowire/nanoribbon, I would suggest to present two simulated STM images, one for narrow and one for wide nanoribbons, and compare the simulated STM images to experimental images.

Point-by-point response to “Article ID: NCOMMS-16-12106-T, Title: Oscillating Edge States in One-dimensional MoS₂ Nanowires”

Reviewers' comments:

Reviewer #1 (Remarks to the Author):

The manuscript reports the preparation of MoS₂ nanowire and nanoribbon on a vicinal Au substrate. The strain induced morphology corrugation and associated edge state modulation were revealed very clearly by using STM/STS combined with first principles calculations. The data and analysis are clean and convincing. However I feel that the reported result is conceptually not new, and not so important to stimulate a general interest. Edge state is commonly observed on surface steps and 1D structures, and strain modulation due to lattice mismatch is very natural. It is not surprising to observe such phenomena. The authors claim that such modulation along 1D wire may tune the 1D transport property, which is of course interesting if the authors can demonstrate it. But that is totally another story. For the present work, I feel it more suitable for a more specific journal such as physical review B.

Response to first referee's comments:

Although edge state are commonly observed on surface steps and 1D structure, this is the first time that edge states **from a single molecular scale MoS₂ wire** with extremely narrow width (0.6 nm width) are studied **experimentally**. We showed convincingly how such ultrathin nanowires can be prepared with a high coverage on Au surface. Due to the extreme width of such nanowires, the property of the MoS₂ nanowire is dictated by its edge states. There are several key features which distinguishes the current work from any previous reports (if any)

1. The **oscillation of edge states** has not been reported thus far. In fact this is true not just for MoS₂, but for any form of semiconductor or metallic nanowires. Our work *reported such oscillation for the first time* in an ultra-narrow nanowire.
2. **The dependence of the conductance oscillation of this edge state on system size** has never been systematically studied and verified, and we did so in this work. We showed that this oscillation is active only when the system size is reduced to a single molecular wire.

3. We have obtained direct atomic visualization of how 2 adjacent nanowires fuse to form 1 twin wire.

Recently, there are lots of interests on the edge states of graphene ribbon (GNR) and TMDs because of their influence on the electronic and magnetic properties of 1-D system¹⁻². There have been many theoretical studies but not enough experimental verifications on whether such single wires can be grown. In this work, we present an atom-resolved STM/AFM and STS study of MoS₂ nanowires, which helps us to understand the relationship between electronic structure and local atomic geometry. Our results are also important for theoretical studies of TMDS nanoribbon structures, since no ultra-narrow MoS₂ nanowires were ever grown by a bottom up method before. It has important relevance to 1-D materials as a whole, especially when reduced to very fine dimensions, where edge states dominate the properties.

References

- 1 Barja, S. *et al.* Charge density wave order in 1D mirror twin boundaries of single-layer MoSe₂. *Nat Phys* **advance online publication** (2016).
- 2 Chu, R.-L. *et al.* Spin-orbit-coupled quantum wires and Majorana fermions on zigzag edges of monolayer transition-metal dichalcogenides. *Physical Review B* **89**, 155317 (2014).

Reviewer #2 (Remarks to the Author):

The manuscript reports the synthesis and characterization of different new MoS₂ nanowires (around 200 nm long) grown by molecular beam epitaxy using gold substrates. The nanowires/nanoribbons grow at the steps of Au (755) surface. In particular, the thinnest nanowire exhibits a structural and electronic modulation due to the edge interaction with the Au substrate. As the nanowires get wider, the modulation vanishes since the interaction with the substrate changes. Different characterization techniques are used to analyze the properties of these MoS₂ nanowires such as STM, NC-AFM, XPS and HREELS. Additionally, the authors have performed first principles calculations to support their findings. The results look correct, interesting, useful to experimentalists and theorists, and could lead to the controlled synthesis of other semiconducting transition metal dichalcogenides (TMDC) nanowires/nanoribbons, similar in structure (trigonal prismatic), such as WS₂, MoSe₂ and WSe₂. I recommend the publication of the manuscript after the authors address the following points:

1. How stable would the MoS₂ nanowires be if these are exposed to air?

Response: Due to the reactive nature of the edges of such ultra-narrow wires, the nanowire will react with oxygen. However, the structural integrity of the wire is kept as we can still observe it under STM. The thicker wires are less reactive than the thinner ones, due to the higher ratio of basal plane atoms to edge atoms.

2. Can the authors comment about the possibility of releasing the nanowires from the gold substrate to use them in other applications? Is this possible?

Response: In this paper, our samples were prepared on the single crystal gold substrate, so it is costly to peel out the MoS₂ nanowires by chemically etching the gold. In future, we will attempt to synthesize the nanowires on Au or Cu foils. Alternatively it is possible to prepare epitaxial gold films on mica surface for the growth of the MoS₂ nanowires, and the ultrathin gold film can be etched when we transferred the wires using the polymer stamp approach, similar to transferring graphene. In this way, the nanowires may be delaminated from the gold substrate. Due to the technical challenge involved in these steps, they are not attempted in the current study which has its focus on the atomic structures of these wires.

3. According to the authors DFT calculations, the suspended thinnest nanowire possesses a band gap of 0.14 eV. Since DFT always underestimates the band gap, the authors should mention this in the text. Can the authors provide a theoretical realistic value of this band gap for the suspended system?

Response:

The referee is correct to point out that in most cases, DFT significantly underestimates bandgaps of semiconducting materials by 50% due to the neglect of strong correlations in XC functionals. However, MoS₂ is an exceptional case that DFT is able to generate reasonably good bandgap. For a monolayer MoS₂, a previous DFT study gave a bandgap around 1.7 eV, agreeing very well with the experimental value of 1.8 eV (Sci. Rep. 4, 3987, 2014). Our DFT calculations yielded 1.68 eV of the bandgap of a monolayer MoS₂ (see figure below for the DOS), perfectly matching previous results.

Fig 1: Density of states of a monolayer MoS₂ from DFT calculations. The bandgap is estimated to be 1.68 eV.

To further address the issue, we also did the so-called DFT+U calculations where the effects of strong correlations are introduced through the Hubbard U. DFT+U method was designed to fix the underestimated bandgap by DFT. We found that for monolayer MoS₂, (unlike other materials) the inclusion of U decreases the bandgap which can be seen from Fig. 2. We therefore concluded that the DFT prediction of 0.14 eV for the thinnest MoS₂ wire we synthesized is reasonable. We have added this comment in supporting file.

Fig. 2: Bandgaps of a monolayer MoS₂ from DFT+U calculations. Note that the Hubbard U is in the unit of eV. The inclusion of U decreases the bandgap.

4. Can the authors comment, in the context of their findings, how their technique could be extended to other semiconducting STMD such as WS₂ and WSe₂ and how the spin orbit coupling, which is greater than in MoS₂, could play a role in new electronic/magnetic properties, when the nanoribbons get wider?

Response: Yes, we think our findings can be applicable to the synthesis of other TMDs nanowires, such as WS₂ and WSe₂. The lattice constant of the MoS₂ ($a=3.183\text{\AA}$) is close to that of WS₂ ($a=3.182\text{\AA}$). Due to the larger spin orbit coupling, the bandgap of the monolayer WS₂ and WSe₂ are normally larger than that of MoS₂ and MoSe₂ respectively. If strains are exerted on the TMDS nanowires by the substrate, the strains will modulate the coupling strengths of the orbital through the structural relaxations, which result in the change of the band gap. According to DFT calculations, the reduction rate of bandgap in WS₂(WSe₂) is slower than that of MoS₂(MoSe₂) under a systematic increase of strains(*Phys. Rev. B88, 195420, 2013*). Furthermore, it was found that the magnetic moment in WS₂ is larger than the that of MoS₂ when the nanoribbons get wider (*Applied Surf.Sci.371, 376, 2016*) . The concept of half-metallicity can be explored in these 1-D

TMD wires. It has been predicted by theoretical calculations that bare zigzag MoS₂ nanoribbons become half-metallic as a result of the (2x1) reconstruction of edge atoms and are semiconductor for minority spins, but metallic for the majority spins (*J.Phys.Chem.C115, 3934,2011*). These effects may be accentuated in other TMDs with larger spin orbit coupling. *Rui-lin Chu* recently predicted from their first-principles calculations that TMD provide a very good platform for Majorana fermions based on its edge states with strong spin-orbit coupling. As a result, the proximity induced superconducting pairing and the associated Majorana fermions can be robust against disorders (see *Rui-lin Chu, "Spin orbit-coupled quantum wires and Majorana fermions on zigzag edges of monolayer TMD", Physical Review B, 89, 155317 (2014)*).

Action:

We have added the following in the discussion section of the paper

"It has been predicted by theoretical calculations that zigzag MoS₂ nanoribbons become half-metallic as a result of the (2×1) reconstruction of edge atoms and are semiconductor for minority spins, but metallic for the majority spins³⁵. Using first-principles calculations²⁰, Chu et. al. predicted that chalcogen-terminated zigzag edges support edge bands with strong Rashba-type spin orbit coupling which are well separated from the bulk bands; the edge modes can be topological although the bulk semiconductor is non-topological. These effects are expected to be accentuated in TMDs with large spin orbit couplings. The potential modulation and charge ordering observed here could result in the peculiar spin texture and coherent spin dynamics for spin manipulation and detection in these wires."

Reviewer #3 (Remarks to the Author):

Mainly with the STM, Authors studied one-dimensional MoS₂ grown at the step edge of Au surface, found a periodical oscillation. The observation is original and interest, although this phenomenon is very similar to 2D moire pattern. the data were collected carefully and analyzed reasonable, the presentation is clear, the conclusions sound right and useful. There are a few points should be considered before publication.

Response to third referee's comments:

1. The treatment of uncertainties. Authors mentioned the period is 4.4 ± 0.2 nm, but I cannot find why is ± 0.2 , please specify it.

Response: The period is measured by analyzing the line section profiles of 30 *dI/dV* maps of MoS₂ nanowires, taking the statistical mean of their period. The uncertainty arises from the standard deviation of this mean period.

2. Line 151,152. "On the other hand, it is noted that a peak with similar position as P2 has been reported on the edges of MoSe₂". I do not understand why authors mentioned a MoSe₂ peak. The materials are different; it is useless to compare two STS peaks from two materials.

Response: Yes, we agree, this sentence has been deleted in the text.

3. Line 168, 169. "This asymmetric behaviour is similar to the moire pattern inversion between the filled and empty electronic states in monolayer MoS₂/Au system." Please give a Ref. article.

Response: Actually, this is a typo error, we have changed this sentence to "This asymmetric behaviour is similar to the moiré pattern inversion between the filled and empty electronic states in monolayer graphene/Ru system (*Phys.Rev.Lett.*100, 056807,2008)." In Gr/Ru system, the STS recorded on the top of the "high" and "low" regions are obviously different due to inhomogeneities in charge distribution due to electron doping from Ru substrate. This is analogous to the observation in our MoS₂ nanowires

4. In this manuscript, Line 118, the edge of MoS₂ was considered to be terminated by mono-S. Do Authors check the edge be terminated by dimer-S?

Response: We did not observe dimer-S on ultranarrow MoS₂ wire. However we do observe a bright-dark alternating pattern along the edge of tooth-saw like nanoribbons as shown in Fig. 3, in which a periodicity is twice the lattice of MoS₂(3.15Å). This 2× period is typical characteristic of S₂ dimer terminated zigzag Mo edge (*Nat Nanotechnol* 2, 53,2007). It is not clear to us at this stage how the terminations differ, but it may be related to fluctuations in chemical potential of S and Mo during growth.

Fig.3 MoS₂ nanoribbons synthesized on Au (100) surface. The clear S₂ dimer terminated Mo edge was shown in (b) marked by the arrow.

5. Based on the STM image and DFT calculation, author show the structures of the nanowire/nanoribbon, I would suggest to present two simulated STM images, one for narrow and one for wide nanoribbons, and compare the simulated STM images to experimental images.

Response: For systems under study, it is difficult to compare the simulated STM images with experiments because for such ultra-thin wires, the STM tip induced local field normally has great effects, while in simulations, STM tip is not included and the STM image is approximated by ground-state surface local DOS. To demonstrate this, we conducted DFT calculations to simulate the STM image of the single wire. Results are shown in Fig. 4. As expected, although we do see the variation of DOS along the wire, but it is hard to compare simulations with experiments. Actually, in this case, in our opinion, it may make more sense to relate the charge redistribution (Fig. 3g in the paper) with experimentally observed STM image. The charge redistribution mainly happens at the contact region between MoS₂ and Au substrate, which does not affect

much the surface local dos, but surely have great effects on electron tunneling from the Au substrate to the STM tip that will be reflected in observed STM images. On the other hand, we fully agree with the referee that it would be nice to be able to directly explain the observed STM images from theory. This will have to be done by more advanced calculation that fully takes into account the STM-tip induced nonequilibrium effects. We plan to do this using the recently proposed steady-state DFT that is good for nonequilibrium quantum systems (Sci. Rep. 5, 15386 (2015)) to address this issue in our future study. We have added more discussions for this in the resubmitted paper.

Fig.4 Simulated STM image of single MoS₂ wire on Au substrate ($V_s = -1.0V$) showing oscillation in the conductance.

REVIEWERS' COMMENTS:

Reviewer #2 (Remarks to the Author):

The authors have addressed the points I raised in my report. In my opinion the paper is suitable for publication in its present form.

Reviewer #3 (Remarks to the Author):

The points raised have been addressed enough; I think the manuscript is ready for publication.